# Multi-Technique Approach for Work Function Exploration of Sc_2_O_3_ Thin Films

**DOI:** 10.3390/nano13081430

**Published:** 2023-04-21

**Authors:** Alessio Mezzi, Eleonora Bolli, Saulius Kaciulis, Alessandro Bellucci, Barbara Paci, Amanda Generosi, Matteo Mastellone, Valerio Serpente, Daniele Maria Trucchi

**Affiliations:** 1Institute for the Study of Nanostructured Materials, ISMN-CNR, Montelibretti, 00010 Rome, Italy; 2Institute of Structure of Matter, DiaTHEMA Lab, ISM-CNR, Montelibretti, 00010 Rome, Italy; 3Institute of Structure of Matter, ISM-CNR, 00133 Rome, Italy

**Keywords:** scandium oxide, work function, XPS, UPS, AFM, XRD, EDXR, electron-beam evaporation

## Abstract

Thin films based on scandium oxide (Sc_2_O_3_) were deposited on silicon substrates to investigate the thickness effect on the reduction of work function. X-ray photoelectron spectroscopy (XPS), X-ray diffraction (XRD), energy dispersive X-ray reflectivity (EDXR), atomic force microscopy (AFM), and ultraviolet photoelectron spectroscopy (UPS) measurements were performed on the films deposited by electron-beam evaporation with different nominal thicknesses (in the range of 2–50 nm) and in multi-layered mixed structures with barium fluoride (BaF_2_) films. The obtained results indicate that non-continuous films are required to minimize the work function (down to 2.7 eV at room temperature), thanks to the formation of surface dipole effects between crystalline islands and substrates, even if the stoichiometry is far from the ideal one (Sc/O = 0.38). Finally, the presence of BaF_2_ in multi-layered films is not beneficial for a further reduction in the work function.

## 1. Introduction

Energy consumption is estimated to have doubled in the last 30 years [1]. Considering the technology advancements, it can be easily predicted that energy demand will increase further in the near future. Currently, a major amount of energy (~85.0%) is supplied by non-renewable energy sources based on the combustion of fossil fuels, including petroleum, natural gas, and coal [2,3]. However, their exploitation is limited and harmful to the Earth’s health, contributing to global warming and climate change. For this reason, renewable energy sources are assuming a prominent role in the production of sustainable and green energy, representing a good solution for the mitigation of climate change, especially in the reduction of carbon emission in the atmosphere [1]. Thus, in the future, it will become highly challenging to design and develop devices able to exploit renewable energy with the highest possible efficiency.

Multiple sources of energy can be relevant to reach these objectives, including solar energy, geothermal energy, wind, tidal, biomass, biofuels, and many others [4]. Among all of them, the most inexhaustible resource in the human time scale is certainly attributable to the sun; the conversion of its radiation energy can be achieved by using various processes, such as photovoltaic, photothermal, and photocatalytic effects. Recently, novel solid-state hybrid conversion systems were proposed and demonstrated [5,6,7], where both photon and thermal energies are exploited in a thermionic–photovoltaic (TIPV) device. This system converts a high thermal flux, such as concentrated sunlight, by exploiting both electrons and photons as energy carriers, which are efficiently generated at a high temperature, giving rise to a higher power density production [8]. To achieve competitive values of conversion efficiency, the materials involved in the TIPV conversion process must have a low work function because it guarantees high emission and the collection of electrons between the two active electrodes of the device, i.e., the cathode and the anode. Basically, the work function is the minimum amount of energy necessary to remove electrons from a given material and is defined as the difference between the Fermi and vacuum levels. When the work function is low, the electrons can be easily extracted from the cathode, consequently generating a higher flux of electrons towards the anode.

The thermionic process is described by the Richardson–Dushman equation, where the emission current density depends on the material work function and temperature [9]. Following this equation, higher emission currents can be achieved by increasing the temperature and/or by reducing the work function. To preserve the performance of the device for long operating times, it is better to work at a temperature as low as possible, avoiding or limiting the loss of material by evaporation; in this sense, it may help to select materials with thermal stability at moderate temperatures (<1000 °C [10]). Moreover, this approach will guarantee the use of dielectric microspacers [11,12], which are able to limit the space–charge conditions and maintain a proper thermal gradient between the two electrodes.

Therefore, the main diffused strategy is to minimize the cathode work function. Furthermore, to guarantee a correct operation and to maximize the output voltage (that is equal to the difference between the work function values of the cathode and the anode), the anode must have a work function even lower than the cathode. Therefore, it appears clear that the selection of suitable materials for both the cathode and the anode becomes crucial to achieve a higher performance of TIPV converters.

A typical strategy used to reduce the values of cathode and anode work functions consists of the deposition of a thin layer of specific compounds on top of the electrodes’ surface. In the recent past, we investigated and tested several materials used as coatings of cathodes or anodes in TIPV converters, such as lanthanum borides, aluminum nitrides, and barium fluorides [13,14]. In a systematic analysis, the structural, morphological, and chemical compositions of thin films were compared with their work functions, which were determined by ultraviolet photoelectron spectroscopy (UPS).

In the present study, we described the analytical methodologies employed for the investigation of scandium oxide with the aim of exploring its potential application as an active layer for TIPV cathodes or anodes. Scandium is the first transition metal of the periodic table, and, at the same time, it is classified as the lightest element of the rare earth group. Despite scandium being a light element, it is characterized by a relatively high melting point and good mechanical properties [15]. In literature, there can be found two values of the Sc_2_O_3_ work function, which are calculated from thermionic emission measurements and dependent on the temperature range: (1) *φ* = 4.23 eV at T = 1300 ÷ 1700 K; and (2) *φ* = 3.66 eV at T = 1700 ÷ 2000 K [15]. It is interesting to observe that for a very thin film (~3 nm) of Sc_2_O_3_ deposited on tungsten, it was measured at a value of ~1.66 eV and further reduced to ~1.50 eV after the addition of BaO, forming the complex Ba-Sc-O [16].

Based on these reported results, we deposited Sc_2_O_3_ thin films on Si substrates using electron beam physical vapor deposition (EB-PVD), which is a relatively simple deposition method where an electron beam is focused on a selected target, and the evaporated material is condensed on the substrate. The thicknesses of the films were varied in the range of 2 ÷ 50 nm to identify the best condition for achieving the minimum work function value and to investigate the correlation between the structural and chemical compositions of the samples and the work function. Finally, the double layers of Sc_2_O_3_/BaF_2_ and BaF_2_/Sc_2_O_3_ on Si substrates were also investigated to detect the possible effects induced by the formation of mixed compounds.

## 2. Materials and Methods

### 2.1. Samples Preparation

Thin films of Sc_2_O_3_ and mixed multilayers of Sc_2_O_3_ and BaF_2_ were deposited by EB-PVD. The evaporation chamber was equipped with a system that enabled the hosting of up to 4 different target materials. Tungsten and copper crucibles were used to load Sc_2_O_3_ powder (purity 99.999%, produced by MaTecK GmbH, Jülich, Germany) and BaF_2_ pellets (purity 99.9%, supplied by Testbourne Ltd., Basingstoke, UK), respectively. The electron beam was controlled by electromagnetic lenses, which precisely impinged the target materials onto the crucible with a spot size smaller than the crucible diameter. The beam hit the target with a sinusoidal shape for the best thermal conditions and homogeneity of the evaporation flux. Single-crystal (<100>) p-type Si substrates in a holding plate were placed at about 20 cm above the crucible system. The native surface oxide on the Si substrates was previously removed using wet chemical etching for 120 s in HF:H_2_O = 1:10, followed by abundant rinsing in deionized water. The base chamber pressure was <3.5 × 10^−7^ mbar before the deposition. The beam accelerating voltage and current were fixed to 8.3 kV/65 mA and 6.7 kV × 10 mA for Sc_2_O_3_ and BaF_2_, respectively, maintaining the constant deposition rate at 0.04 nm/s. An operative pressure of ~5.0 × 10^−6^ mbar was reached during the deposition. An in situ quartz microbalance, connected to an automatic shutter valve, was used to monitor the film deposition rate and to obtain the desired thicknesses. The film thicknesses of Sc_2_O_3_ on Si were 2, 3, 4, 5, 10, 30, and 50 nm, and these samples were labeled as SCO-2, SCO-3, SCO-4, SCO-5, SCO-10, SCO-30, and SCO-50, respectively.

### 2.2. Characterization Techniques

XPS and UPS measurements were performed using an ESCALAB 250Xi spectrometer (Thermo Fisher Scientific Ltd., East Grinstead, UK) equipped with a monochromatic Al Kα source (hν = 1486.7 eV) for XPS, a He lamp (He I-hν = 21.2 eV and He II-hν = 40.8 eV lines) for UPS, and a six-channeltron detection system. All samples were fixed on the sample holder by metallic clips, and the BE scale was calibrated by positioning the adventitious carbon peak C 1s at BE = 285.0 eV. The XPS and UPS spectra were acquired at constant pass energy of 50 eV and 5 eV, respectively. In order to eliminate the contribution of the spectrometer work function, the UPS spectra were acquired by applying different bias voltages to the samples, following the method described elsewhere [13,14]. The spectra were collected and processed using the Avantage v.5.9 software (Thermo Fisher Scientific Ltd., UK).

X-ray diffraction measurements were performed by means of a Panalytical-Empyrean diffractometer equipped with a ceramic Cu-anode X-ray source and a PixC’el 3D detector working in linear mode. Patterns were acquired in a Bragg Brentano configuration, with divergent slits (1/4°–1/2°) focusing onto the sample positioned on a flat stage sample holder. XRD patterns were collected in the 20° < 2θ < 70° angular range (step size [°2θ] = 0.0260, scan type continuous, step time [s] = 800). The generator was kept at a current intensity of 45 mA and a voltage of 40 kV.

Energy-dispersive X-ray reflectivity (EDXR) measurements were performed by means of an unconventional in-house-developed energy-dispersive X-ray reflectometer (Patent No. RM 93 A000410, 1993, R. Felici, F. Cilloco, R. Caminiti, C. Sadun, and V. Rossi Albertini, Italy). EDXR patterns were obtained, due to the non-symmetric configuration of the apparatus, maximizing the totally reflected radiation. A polychromatic incident radiation produced by a W-anode (energy range 10–50 keV) was used as the probe, with no monochromator needed. A solid-state high-purity Ge single crystal (ORTEC) accomplished the detection, as it was sensitive not only to the number of incoming photons but to their energy too. The detector was kept at cryogenic temperatures via an electro-mechanical cooler. The samples were mounted into an appropriate sample holder that was transparent to X-rays, and the incident/reflected pathway was focused using W rectangular slits (horizontal aperture: 1000 µm; vertical aperture 40 µm). An Al (2 mm) filter was used to reduce the W-anode fluorescence lines.

An OmegaScope platform integrated into a LabRAM HR Evolution Raman microscope (HORIBA Ltd., Kyoto, Japan) was used for morphological and roughness parameters information. AFM imaging was operated in AC mode on square areas of 10 × 10 μm^2^ using a silicon pyramidal tip (MikroMasch HQ: NSC14/Al BS, Wetzlar, Germany), with a tip radius of 8 nm. The scan rate was fixed at 0.8 Hz. AFM images were recorded with a 500 × 500 pixel definition, corresponding to 20 nm of spatial resolution. The AIST-NT SPM control software was used to acquire and analyze all the AFM images.

## 3. Results and Discussion

### 3.1. X-ray Photoelectron Spectroscopy (XPS)

The chemical compositions of the samples were investigated by XPS, providing information on the elemental composition and oxidation states of the elements in the first layers of the investigated materials (information depth < 10 nm) [17]. Therefore, the XPS technique was very useful for the investigation of the Sc, O, and C spectra in Sc_2_O_3_ thin films. The XPS quantitative analysis evidenced a remarkable amount (20 ÷ 40 at.%) of C surface contamination, which was almost completely removed after 30 s of Ar^+^ ion sputtering at 1 keV. In the case of ultra-thin films (2 ÷ 5 nm), the signals from the Si substrate were also registered, due the analysis depth being higher than the film thickness, even if the presence of non-homogeneous Sc_2_O_3_ films could not be excluded in principle. In Figure 1, the atomic concentration ratios of Si/Sc and Sc/O, determined after the surface cleaning, were plotted as functions of the film thickness. As it can be seen, the Si contribution became negligible for thicknesses over 10 nm, whereas the thinner films were characterized by a very low Sc/O atomic ratio, due to the presence of ScO(OH) formed in the air.

A Sc 2p spectrum acquired for the 50 nm thin film is shown in Figure 2. This photoemission signal was composed of a spin–orbit doublet (Sc 2p_3/2_ and Sc 2p_1/2_) separated in energy by 4.5 eV.

All the investigated samples were characterized by the same Sc 2p signal, where the Sc 2p_3/2_ peak was positioned at BE = 402.2 ± 0.1 eV. This value was associated with Sc_2_O_3_; however, the presence of the insoluble hydrous oxide ScO(OH) or the sub-oxide Sc_x_O_y_, formed after the air exposure on the film surface, could not be excluded [18]. This aspect was enforced by the XPS quantitative analysis shown in Figure 1, which highlights the fact that the Sc/O atomic ratio was never stoichiometric (Sc:O = 2:3), showing a small excess of oxygen (Sc/O ~1.0, rather than Sc/O ~0.6). The presence of ScO(OH) was established from the examination of the O 1 s signal. As it can be seen in the Figure 3, there were two contributions in the O 1 s signal (black line), positioned at BE = 530.1 and 531.8 eV, that could be assigned to Sc_2_O_3_ and ScO(OH), respectively. A third peak at BE = 533.0 eV, was attributed to adsorbed water (from air) or, in the case of thinner films, to native oxide SiO_2_ on the Si substrate. Before cleaning the surface using ion sputtering, the intensity of the ScO(OH) peak was relatively high, but it was considerably reduced after the surface cleaning, indicating that it was formed on the film surface after exposure to air.

This result was also confirmed by the XPS depth profile (shown in Figure 4), which consisted of cyclic ion sputtering alternated with spectra acquisition. Once the excess of oxygen was removed, the stoichiometric value of Sc_2_O_3_ was achieved. However, since the ion sputtering was a destructive process, there was the possibility to create artefacts. Anyway, the profile concentration of Sc and O resulted almost constant down to the interface with the Si substrate.

In the case of the Sc_2_O_3_ film covered with 2 nm of barium fluoride, there were no substantial differences from the compositional point of view. Below the barium fluoride layer, the chemical composition of Sc_2_O_3_ was stoichiometric without any formation of the ScO(OH) species. It seemed that barium fluoride acted only as a protective capping layer. Effectively, in this case, a small excess of oxygen was attributed to the partial oxidation of BaF_2_, as it was already observed in our previous work [14]. Moreover, as shown in the depth profile (Figure 5), no diffusion of Ba and F occurred in the Sc_2_O_3_ film with the formation of the neat interface.

Instead, when the BaF_2_ layer was located between the Sc_2_O_3_ film and the Si substrate, the XPS depth profile evidenced the diffusion of Ba and F. In fact, the signals of Ba 3d and F 1s already emerged on the sample surface and remained throughout the whole film, down to the substrate (see Figure 6). In this case, the atomic ratio Ba/F = 0.4 on the surface was similar to the stoichiometric one, whereas this ratio changed to almost Ba/F = 1 when going towards the Si substrate. Interestingly, the atomic ratio Sc/O was constant along the whole film.

### 3.2. X-ray Diffraction (XRD) and Energy Dispersive X-ray Reflectivity (EDXR)

The structure of the Sc_2_O_3_ thin films was investigated using XRD. Figure 7 shows the acquired XRD patterns of the samples SCO-2, SCO-10, and SCO-30, respectively. For the SCO-2 sample, the only diffraction peak of Sc_2_O_3_ (211) was identified at 2θ = 22.12°, suggesting a monocrystalline growth of the film; reversely, the 20 nm and 30 nm films exhibited a polycrystalline scandium oxide structure, with no preferential orientation, labeled according to the Sc_2_O_3_ ICDD card nr: 00-005-0629 [crystal system: cubic, space group: Ia3, space group number: 206] [19]. However, no further crystallographic features of other oxides were found, confirming the formation of only a few layers of amorphous ScO(OH) on the top of the films.

Subsequently, the samples were morphologically characterized using the EDXR technique, and the results are shown in Figure 8. The sample SCO-2 did not show Kiessig contributions (oscillations due to constructive interference between the reflection at the film/substrate interface), due to the film thickness (nominally 2 nm) being below the detection limit of this technique. Nevertheless, the Fresnel reflectivity curve (typical of bulk material) was fitted, and the so-obtained roughness value σ was 1.0 nm. Reversely, samples SCO-10 and SCO-30 exhibited the characteristic oscillating EDXR pattern, thus allowing for the thickness and the combined surface/interface roughness to be calculated using the Parratt fitting procedure (red line in Figure 8). The so-obtained thicknesses (11.7 and 23.0 nm, respectively) were in good agreement with nominal values, and the calculated roughness results were equal to 0.5 and 0.8 nm.

Concerning the multilayer structures, the sample with 2 nm of BaF_2_, deposited on the 20 nm Sc_2_O_3_ film, showed the same diffraction pattern as Sc_2_O_3_ (see Figure 9), with the addition of a peak at 2θ = 37°, due to the crystallographic reflection of BaF_2_ (130) [ICDD nr. 00-034-0200, crystal system: orthorhombic, space group: Pnam, space group number: 62]. However, this peak disappeared in the sample with the BaF_2_ layer positioned between the Sc_2_O_3_ film and the Si substrate. Such a result might be caused by the instability of the metastable phase obtained during the deposition process. This hypothesis was fully in agreement with the XPS depth profile, which showed the diffusion of Sc, O, Ba, and F atoms throughout both films, forming only crystalline Sc_2_O_3_ structures and amorphous mixed compounds based on Ba_x_F_y_.

Subsequently, the same experimental EDXR procedure was adopted to obtain morphological information for multilayered samples of BaF_2_/Sc_2_O_3_ (2 nm/20 nm) and Sc_2_O_3_/BaF_2_ (2 nm/20 nm). In BaF_2_ and Sc_2_O_3_, the theoretical scattering length densities were very similar (3.460 E^−5^ A^−2^ and 3.276 E^−5^ A^−2^, respectively). Therefore, these multilayers were expected to behave as the single films (no interface being detected) from the X-ray reflectivity point of view.

Indeed, as reported in Figure 10, in the sample Sc_2_O_3_/BaF_2_, only a single layer was observed with a thickness of 18.5 nm, calculated using the Parratt fitting procedure (red line), and close to the nominal value (22 nm). Conversely, no Kiessig fringes were observed for the sample BaF_2_/Sc_2_O_3_. The reason for the absence of constructive interference of reflected X-rays was probably related to the fact that the BaF_2_ layer was directly deposited on top of the Si substrate (differently from all other samples, where the scandium oxide was directly deposited on the Si [100]); therefore, it might be too rough or inhomogeneous for a coherent scattering.

### 3.3. Atomic Force Microscopy (AFM)

The topographies of the samples SCO-2 and SCO-10 were investigated using AFM to evaluate the homogeneity of the films. Figure 11 shows the maps obtained for the two samples. As expected, their structures were different: in the sample SCO-2 (Figure 11a), the coating was inhomogeneous, with the presence of some discontinuities (i.e., lack of material) between different islands. The roughness parameters, Ra= 2.0 nm and Rms= 2.6 nm, agreed with the values calculated from EDXR measurements. Comparable values of Ra and Rms were found for SCO-3, SCO-4, and SCO-5 (not shown here). After increasing the nominal thickness above 10 nm, the films became continuous (Figure 11b), characterized by a homogeneous growth on the substrate, with Ra= 1.1 nm and Rms= 1.4 nm. These results confirmed those obtained from the XPS and XRD measurements. Thus, according to the evolution shown by AFM, i.e., the formation of a continuous film as a function of the deposition time (i.e., thickness), the most proper hypothesis was that a Volmer–Weber growth occurred [20] with the creation of separate clusters of crystallites, which tend to grow an epitaxial multi-layer film on the silicon surface.

### 3.4. Ultra-Violet Photoelectron Spectroscopy (UPS)

The UPS is a powerful tool for the investigation of the valence band electronic structure in solids [21]. It is a surface-sensitive technique because the excitation process involves the extraction of photoelectrons with low kinetic energy, which come out from the first monolayers of the investigated material (1 ÷ 2 nm). In the last decade, we used a methodology of UPS investigation, which addressed to the determination of the work function for materials applied in the thermionic field. This method consists of the acquisition of valence band spectra by using He I photons of hν = 21.2 eV. He I spectra can be divided into three regions [21]: (1) Fermi level; (2) valence band structure; and (3) inelastically scattered electrons. The peculiarity of the last region is the achievement of a drastic drop in the signal due to the limit of minimum electron energy required to reach the vacuum level. This cut-off feature makes UPS an exclusive tool for measuring the materials’ work function. Another technique that can be employed for this purpose is Kelvin probe spectroscopy; however, UPS is considered more reliable for absolute work function determination [22].

The work function (*φ*) can be calculated from the following formula (1) after the measurement of the cut-off energy:*φ* = hν (HeI) − BE_cut-off_(1)

In principle, this method for the work function calculation cannot be applied when the material work function value is lower than the value of the spectrometer. When it happens, the sample is typically biased; the best practice is to apply different bias voltages (*V*) and to plot the graph of E_cut-off_ vs. V^1/2^, whose linearity is described by the classical Schottky model for a potential barrier [23]. By extrapolating the cut-off value to *V* = 0, it is possible to determine the real work function value. As shown in Figure 12, the application of bias voltage permits the override of the spectrometer work function and the measuring of the real cut-off energy.

The obtained values of the work function for all the investigated samples are plotted in Figure 13, including the double layer samples of BaF_2_/Sc_2_O_3_ and Sc_2_O_3_/BaF_2_. The black points indicate the *φ* values determined for Sc_2_O_3_ films deposited on the Si substrates. As it can be noted, the *φ* value was constant (~4.2 eV) when the thickness was above 10 nm. Below this thickness, the *φ* value decreased and reached the minimum of 2.7 eV at 2 nm. The red line traced to interpolate the black points revealed a rise in the work function, which is quite similar to the work function curve determined for the BaF_2_ thin films [14], and we hypothesize that the formation of dipoles between the coating and the substrate increases when the surface coverage is not complete. This behavior can be explained by considering the Langmuir–Gurney model [24,25], which predicts that the charge transfer, responsible for the reduction in the work function value, is maximized when the surface coverage is almost 70%. Generally, the typical evolution of the work function presents a local minimum at the optimized value of coverage, which, in our case, corresponded to the thickness of 2 nm, representing the best performance. It should be noted that a similar value was obtained for the sample Sc_2_O_3_/BaF_2_ (2/20 nm), whereas for the sample BaF_2_/Sc_2_O_3_ (2/20 nm), it was close to the values in the samples of Sc_2_O_3_, with a thickness of 20 nm. This finding clearly indicated that the overlayer of BaF_2_ did not help to reduce the work function of scandium oxide, even if the diffusions of Ba and F were present. On the other hand, the *ϕ* values close to the ones previously reported for mixed BaO-Sc_2_O_3_ systems (1.5 ÷ 1.7 eV) [15] were not obtained. It means that in our case, the multilayer approach was not useful, and a study of film deposition from prepared mixed powders should be performed to investigate a possible reduction in *ϕ* down to the values below 2.7 eV.

## 4. Conclusions

Thin films of scandium oxide were deposited on Si substrates by electron beam evaporation to investigate the thickness effect on the work function reduction. The obtained results revealed two different situations: when the sample thickness was higher than 10 nm, the films were polycrystalline, homogenous, and almost stoichiometric. If the thickness was lower than 10 nm, the film was inhomogeneous and composed of crystalline islands. This second case corresponded to the minimization of the work function down to 2.7 eV for the sample with 2 nm of nominal thickness. With the aim of further reducing the work function, the addition of a BaF_2_ layer was investigated, but it was found that its presence did not decrease the value of the work function, regardless of if it was placed on the top or beneath the Sc_2_O_3_ film. We conclude that the required approach for a future investigation will be the deposition of ultra-thin films from the mixed powders of Ba-O-Sc.

## Figures and Tables

**Figure 1 nanomaterials-13-01430-f001:**
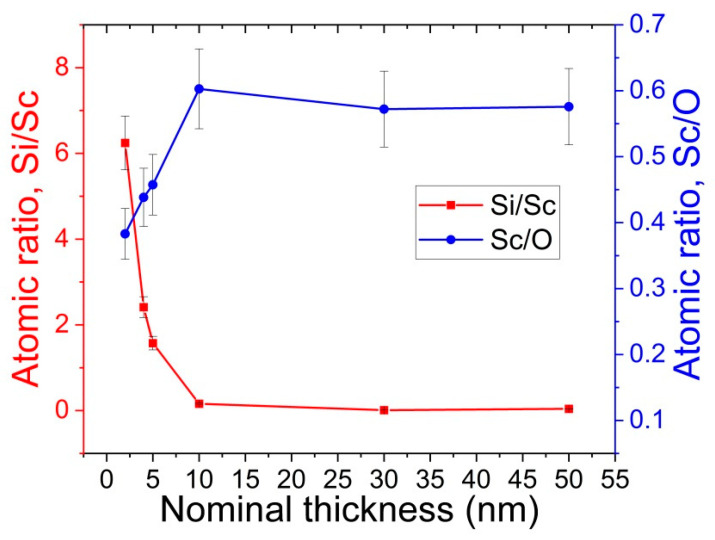
Plot of the Si/Sc and Sc/O ratios vs. nominal thickness.

**Figure 2 nanomaterials-13-01430-f002:**
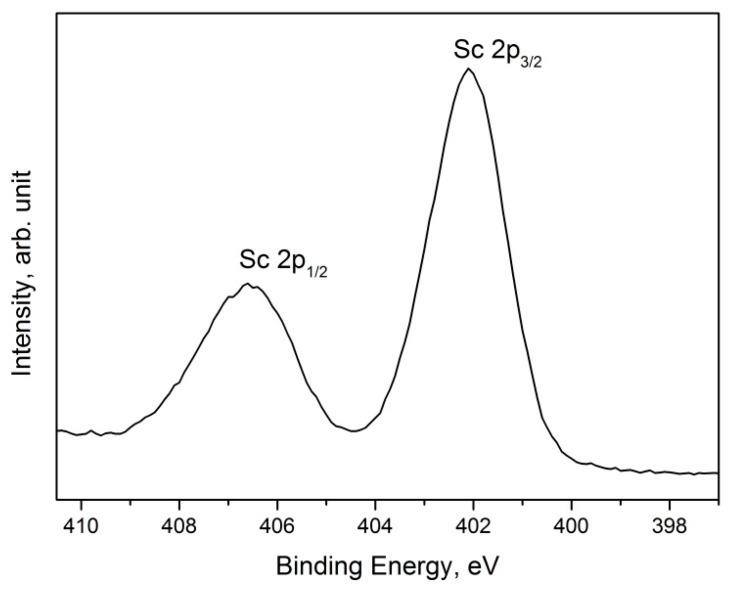
Sc 2p spectrum composed of a typical doublet Sc2p_3/2_–Sc2p_1/2_.

**Figure 3 nanomaterials-13-01430-f003:**
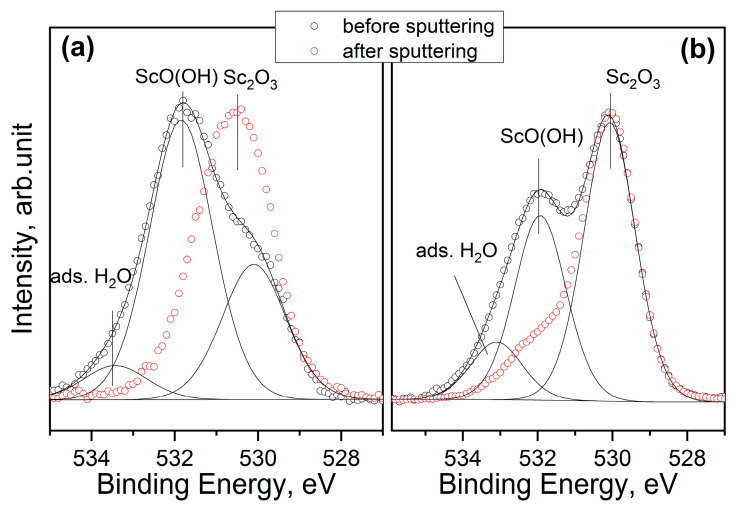
Comparison of O1s spectra before and after 30 s of ion sputtering of the samples SCO-2 (**a**) and SCO-30 (**b**).

**Figure 4 nanomaterials-13-01430-f004:**
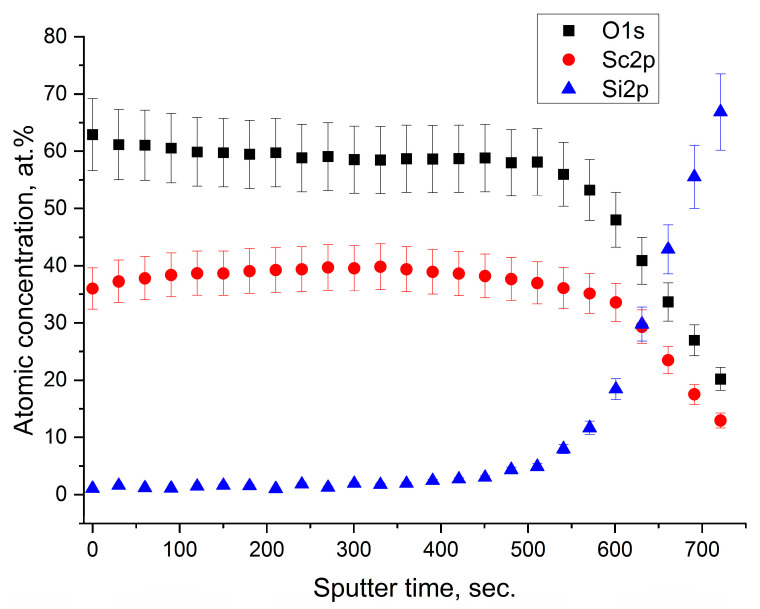
XPS depth profile of the sample SCO-30.

**Figure 5 nanomaterials-13-01430-f005:**
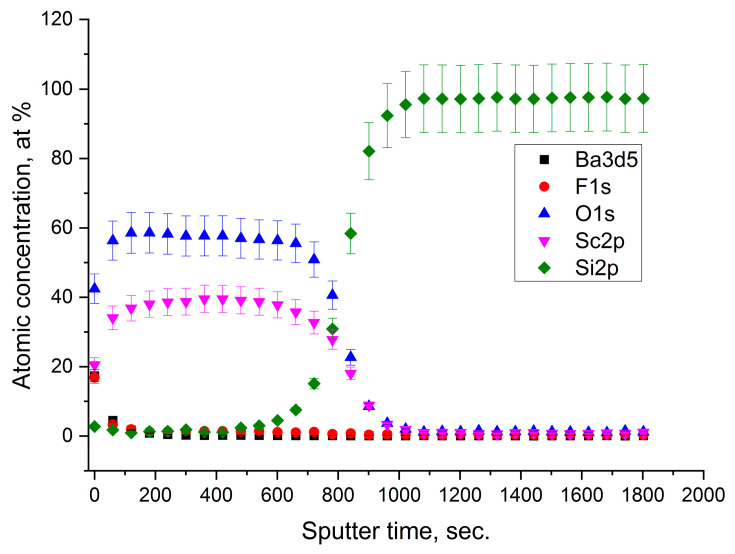
XPS depth profile of the sample with the BaF_2_ thin film on Sc_2_O_3_.

**Figure 6 nanomaterials-13-01430-f006:**
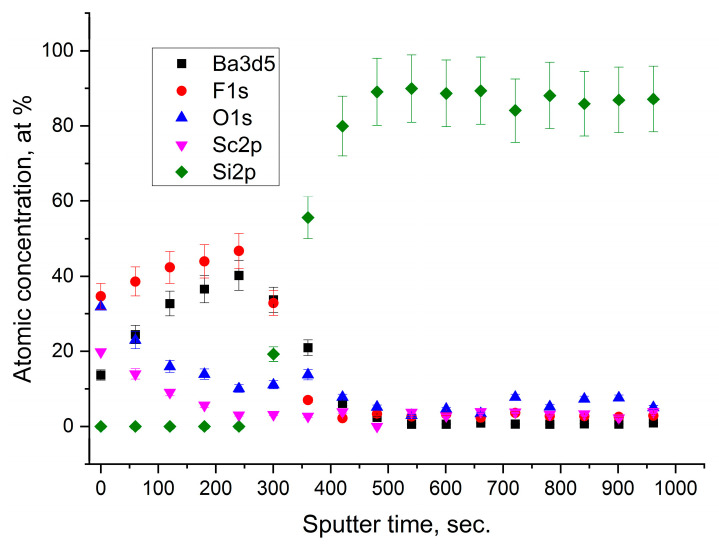
XPS depth profile of the sample with the BaF_2_ thin film between the Sc_2_O_3_ and the Si substrate.

**Figure 7 nanomaterials-13-01430-f007:**
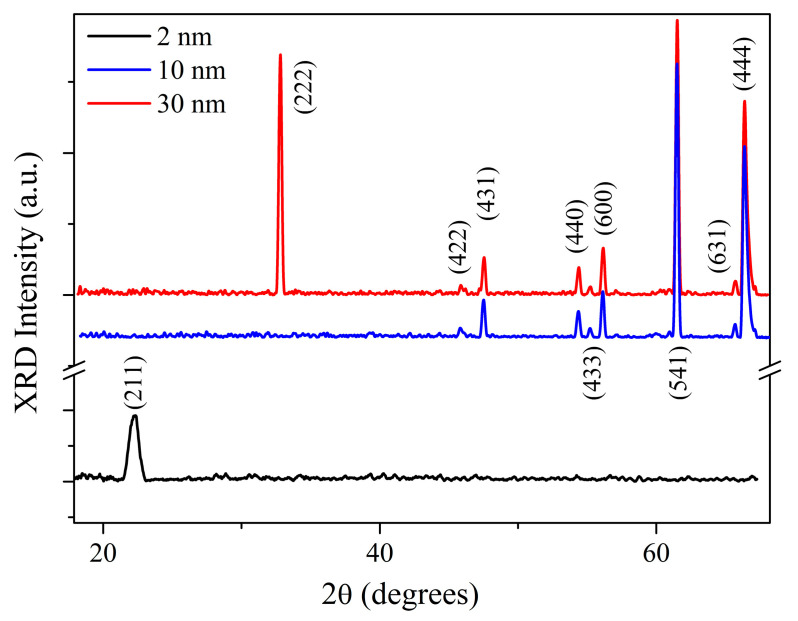
XRD patterns of scandium oxide films.

**Figure 8 nanomaterials-13-01430-f008:**
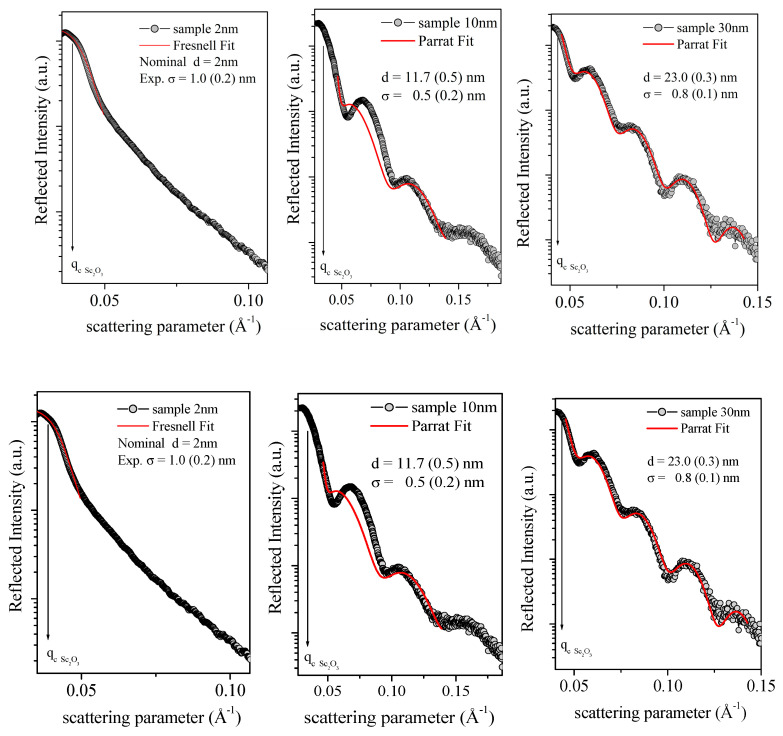
EDXR patterns (black dots) and their fittings (red line) of the samples SCO-2, SCO-10, and SCO-30. Black arrows indicate the critical scattering value for each sample.

**Figure 9 nanomaterials-13-01430-f009:**
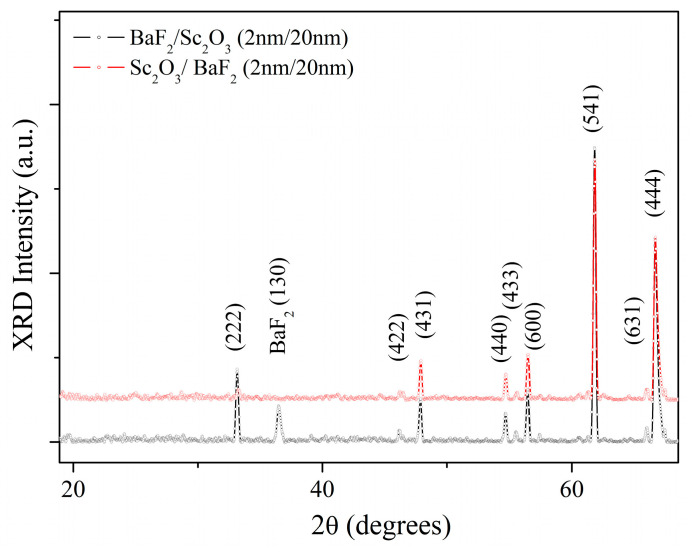
XRD patterns of the BaF_2_/Sc_2_O_3_ and Sc_2_O_3_/BaF_2_ samples.

**Figure 10 nanomaterials-13-01430-f010:**
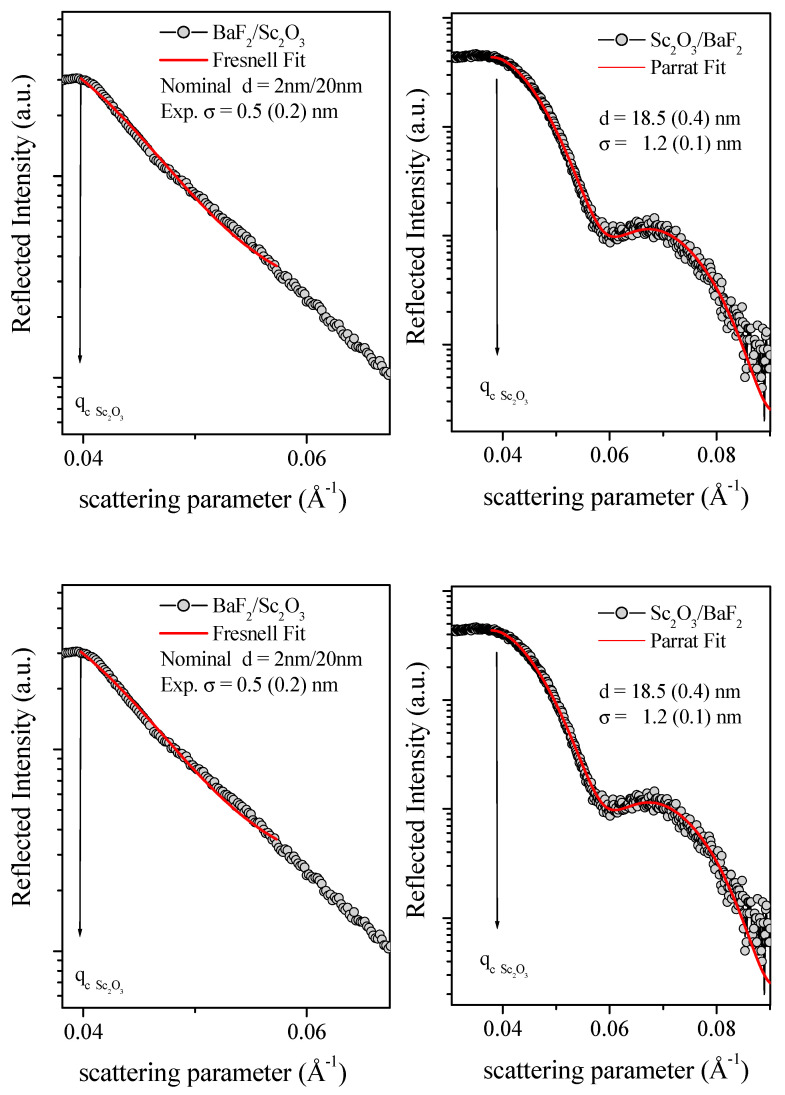
EDXR patterns (black dots) and fittings (red line) of the samples BaF_2_/Sc_2_O_3_ (left side) and Sc_2_O_3_/BaF_2_ (right side). Black arrows indicate the critical scattering value for each sample.

**Figure 11 nanomaterials-13-01430-f011:**
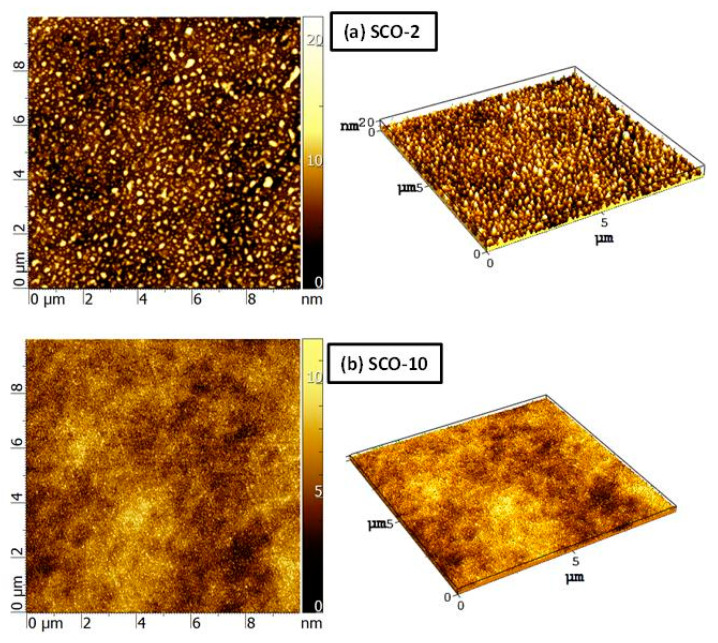
AFM images of the SCO-2 (**a**) and SCO-10 (**b**) samples.

**Figure 12 nanomaterials-13-01430-f012:**
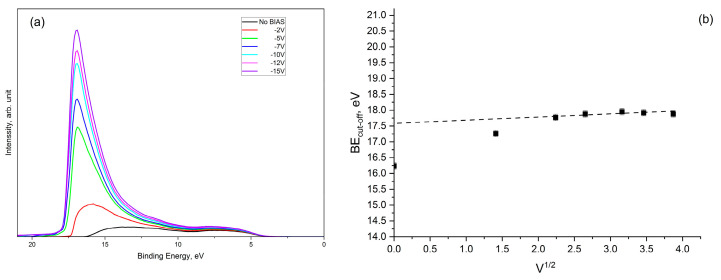
UPS spectra (He I) acquired by applying different bias voltages to the sample (**a**) and the plot of BE_cutoff_ vs. ${\left|V\right|}^{1/2}$ (**b**).

**Figure 13 nanomaterials-13-01430-f013:**
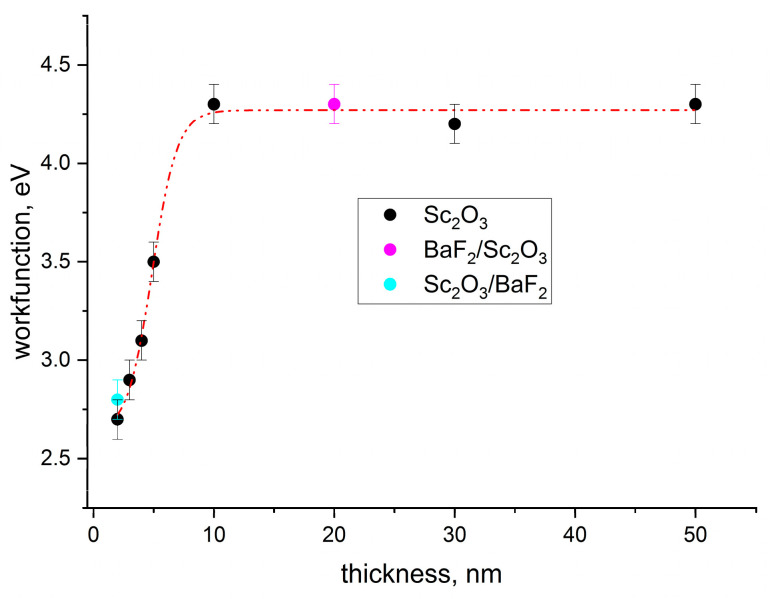
Plot of the work function values as a function of the thickness and the interpolation curve (red line) for both the Sc_2_O_3_ and the double-layer (with BaF_2_) films.

## Data Availability

The data supporting the findings of this study are available from the corresponding author upon reasonable request.

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
