# Peer review of "Multi-Technique Approach for Work Function Exploration of Sc2O3 Thin Films"

_nanomaterials, 2023, doi:10.3390/nano13081430_

Round 1

Reviewer 1 Report

The manuscript titled by “Multi-technique approach for work function exploration of 2 Sc2O3 thin films” report a comprehensive survey on Sc2O3 thin film grown on Si substrate. The authors report an investigation of the thickness effect on the reduction of work function. X-ray photoelectron spectroscopy (XPS), X-ray diffraction (XRD), energy dispersive X-ray reflectivity (EDXR), atomic force microscopy (AFM), and ultraviolet photoelectron spectroscopy (UPS) measurements were carried out the films deposited by electron-beam evaporation with different nominal thicknesses (in the range 14 of 2 - 50 nm) and in multi-layered mixed structures with barium fluoride (BaF2) films.

The manuscript is written well and the experimental data seems reasonable and informative. The authors aim to understand the thickness effect on the reduction of work function. However, the experimental results only show that the inhomogeneous and the formation of crystalline islands might be the key to understand the thickness effect. Such conclusion is not very convincing and substantial, it makes this work less attractive.  The authors may need clear picture or argument regarding the physics of the reduction of work function.  

Some minor comments are listed below, the authors might need consider when they resubmit the paper.

11   What is the surface condition of pristine Si substrate?

22.  Is the reduction of work function observed in other substrates except Si or tungsten?    What is about the other rear earth oxide?

33.  Does the existence of hydrous oxide ScO(OH) affect the work function? Does the ion sputtering have effect on the work function?

44.     XRD pattern of 2nm scandium oxide films is quite unique.  Is 2 nm film amorphous? Sc2O3/ BaF2 (2nm/20nm) is totally different from that of 2nm Sc2O3. Why?

Reviewer 2 Report

In their paper "Multi-technique approach for work function exploration of Sc2O3 + BaF2 thin films", Alessio Mezzi and co-workers show results on the morphology, composition, and crystallanity of Sc2O3 thin films grown by electron beam evaporation. They find that for thin films the growth is discontinuous and that the contact plane is different from that of 10 nm and ticker films. Finally, the work function is determined using UPS measurements and the influence of BaF2 on it is evaluated.

The studied films could find application as electrodes in TIPV devices, which could make the work interesting for the readers of "nanomaterials". However, there are quite a few shortcomings. For this reason, I recommend that the paper should only be published after a major revision.

In more detail:

1. The paper lacks error bars! Without these, all statements about stoichiometric ratios are not substantiated. Some of the experimental results seem to be over-interpreted or do not support the authors' claims.

2. It is not clear whether the samples are examined with or without breaking the vacuum after electron-beam deposition. Therefore, it is not clear whether the discontinuous films form due to Vollmer-Weber growth or whether a wetting layer forms in vacuum but then breaks up into clusters because of contact with air and water vapor. Do the islands/clusters change their size in time, which might hint to Ostwald-ripening?

3. line 63+64: It is not clear what the authors mean. Please rephrase.

4. line 136: "tension of 40 kV"? Do the authors mean "voltage"?

5. Figure 1: Error bars are missing.

6. Figure 2 and line 185: The authors claim that the 2p3/2 peak is at 402.2 eV. Looking at the figure, the peak appears to be at a lower BE than 402 eV. Please explain or correct.

7. line 188: "in Fig. 1, that highlights the fact that the Sc/O atomic ratio was never stoichiometric ...": as long as no error bars or confidence intervals are given, one cannot conclude that.

8. Line 204: The authors claim that "One the excess of oxygen was removed, the stoichiometric value of Sc2O3 was achieved." First, this cannot be concluded sind error bars in Figure 4 are missing. Secondly, the sputter time in Figure 4 stops although the O1s signal is still decreasing.

9. Figures 4, 5, and 6: Error bars are missing.

10. Figure 7: The authors should explain, why the (211) peak is only visible for the very thin film, and why it disappears for thicker films. Has something similar been observed in the literature?

11. Line 307: The authors should make it clearer what they mean by "holes". Holes in the underlying layer, etched away by Sc2O3? Or do they want to describe a Vollmer-Weber growth?

12. Line 308: Since the authors provide both values, Ra and Rms, they should state the difference between these values and how they ar relate to the roughness determined by the other experimental methods.

13. Line 317: There is a missing "t" in "Ultra-violet".

14. Figure 12: The authors should spend a few more words on why only the BE-values for large V^(1/2) are used for extrapolation.

15. Figure 13: Error bars are missing.

16. Line 372: The authors claim that "If the thickness is lower than 10 nm, the film is inhomogeneous and composed of crystalline islands". They have only measured a film with 2 nm and a film with 10 nm thickness, so this statement is not substantiated.

17. References: The authors should take care to use the correct journal abbreviations, e.g. "Appl. Surf. Sci.", not "Appl. Surf. Scie." in Ref 20.

Round 2

Reviewer 2 Report

The authors have addressed most of the comments from my initial report. However, a few questions still remain open.

1. Is the vacuum broken when samples are moved to, e.g., the XPS characterization system?

2. Line 185: The authors still claim that the Sc 2p3/2 peak is at 402.2 eV. Figure 2 clearly shows that the peak is below 402 eV. Please correct!

3. Line 313: How can the authors be sure that the growth is epitaxial?

4. Line 314: please reformulate the basic reason for VW growth.

Therefore, I recommend that the paper should be published after minor revision.
